# Depth-Based Dynamic Sampling of Neural Radiation Fields

**Jie Wang [1,2], Jiangjian Xiao [1,2], Xiaolu Zhang [2,*], Xiaolin Xu [1,2], Tianxing Jin [1,2] and Zhijia Jin [1,2]**

1   Faculty of Electrical Engineering and Computer Science, Ningbo University, Ningbo 315201, China
2   Ningbo Institute of Industrial Technology, Chinese Academy of Sciences, Ningbo 315201, China
*   Correspondence: zhangxiaolu@nimte.ac.cn

**Abstract:** Although the NeRF approach can achieve outstanding view synthesis, it is limited in practical use because it requires many views (hundreds) for training. With only a few input views, the Depth-DYN NeRF that we propose can accurately match the shape. First, we adopted the ip_basic depth-completion method, which can recover the complete depth map from sparse radar depth data. Then, we further designed the Depth-DYN MLP network architecture, which uses a dense depth prior to constraining the NeRF optimization and combines the depthloss to supervise the Depth-DYN MLP network. When compared to the color-only supervised-based NeRF, the Depth-DYN MLP network can better recover the geometric structure of the model and reduce the appearance of shadows. To further ensure that the depth depicted along the rays intersecting these 3D points is close to the measured depth, we dynamically modified the sample space based on the depth of each pixel point. Depth-DYN NeRF considerably outperforms depth NeRF and other sparse view versions when there are a few input views. Using only 10–20 photos to render high-quality images on the new view, our strategy was tested and confirmed on a variety of benchmark datasets. Compared with NeRF, we obtained better image quality (NeRF average at 22.47 dB vs. our 27.296 dB).

**Keywords:** NeRF; scene representation; view synthesis; image-based rendering; volume rendering

## 1. Introduction

Graphically realistic rendering has a wide range of applications, including virtual reality, autonomous driving, 3D capture, and even 3D visualization; however, there are significant challenges in the photo-level rendering of the display world from any viewpoint. Traditional modeling approaches, while yielding high-quality scene geometry, are often too costly and time consuming. Therefore, researchers have developed image-based rendering (IBR), which combines scene-based geometric modeling with image-based view interpolation [1–3]. Despite the great progress, the method has certain problems, as, for some environments, not only do realistic and complex scenes have to be portrayed, but also light and shadow changes and visual angles. To overcome these limitations, the rise of the neural radiance field (NeRF) technique [4] has enabled closer human interaction with the scene and has taken implicit modeling to a new level, modeling the scene as a continuous volumetric field parametrized using a neural network, using MLP to minimize the loss of all observed views in the real. The new view synthesis of the scene produces stunning results.

Although NeRF is able to synthesize photo-quality images with a complex geometry and appearance, NeRF still has many problems: first, the NeRF method requires a huge number of images, often hundreds of images for a "classroom" size scene. As shown in Figure 1, NeRF does not work well for scenes with only a few dozens of images, and the rendering results have large blurring areas. This is because NeRF relies purely on RGB values to determine the correspondence between images, and only a sufficient number of images can overcome this blurring generation to achieve high-quality visual effect. In the case of significantly reduced images, an NeRF technique that relies on color supervision alone is no longer able to construct fine 3D scenes. Secondly, the surface of the 3D model

obtained using NeRF with a small number of inputs is rough. Because NeRF does not intersect rays with the surface of an object at a single point, but rather represents the probability of a series of points, the rendering process of the model relies on the RGB values of the image. Therefore, when only a small number of input images are available, the difference between the rendered image and the training image view is too large to enable the NeRF method to locate the exact intersection of the camera rays with the scene geometry and to show a blurred rendering (Figure 2). Thirdly, the NeRF method requires nearly 200 forward predictions of the MLP depth model for each pixel while producing raw images. Despite the small size of a single computation, the pixel-by-pixel computation to complete the rendering of the whole image requires a significant computational cost. Secondly, the training time required by NeRF for each scene is also slow due to the large number of sampling points required for each pixel point to determine its spatial location.

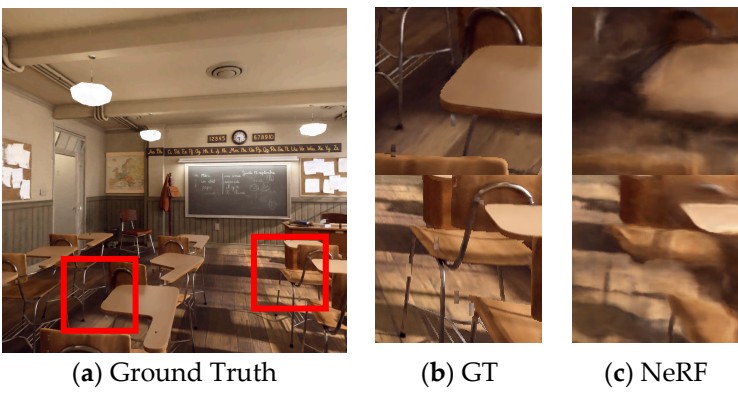

(**a**) Ground Truth          (**b**) GT          (**c**) NeRF

**Figure 1.** Local images of NeRF reconstruction effect on 11 images.

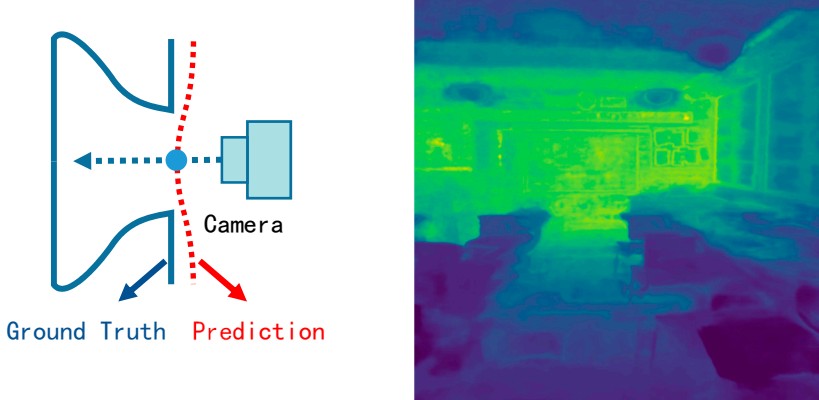

**Figure 2.** Geometric structure constructed by NeRF with a small number of input images (first from left) and the visualized depth information (second from left).

In this paper, we propose the use of a Dynamic Depth Radiation Field (Depth-DYN NeRF), a new implicit representation that can be used for fast and high-quality free-viewpoint rendering. Instead of only modeling space in terms of point positions and orientations, Depth-DYN NeRF constrains the implicit function of NeRF by truncating the symbolic distance function. Specifically, we construct the truncated symbolic distance function based on the depth data of each pixel point and use the multilayer perceptron (MLP) to build the 3D model only in the parts with content, gradually learning Depth-DYN NeRF from coarsely to subtly, and building voxels that do not contain scene information, so that the network focuses on learning the volumetric regions with scene content for implicit function learning. We determined the sampling range of sampling points based on the depth data of each pixel point to avoid blank sampling points as much as possible, which can remove a large number of empty voxels without scene content and greatly accelerate

the rendering at the time of inference. In the rendering stage, as shown in Figure 3, our specific approach is used for processing the depth information obtained using a light detection and ranging (LiDAR) or depth camera through a depth-completion network to make the depth information more accurate and complete. We constructed a point cloud model through the depth forward projection of a small number of images and obtained the depth information of the corresponding pose on the point cloud model and completed the depth information using a depth-completion algorithm. We evaluated our method on the dataset constructed using DoNerf [5], which only uses a small number of input images, and extensive experiments have shown that our method outperforms most of the current mainstream algorithms with a small number of image inputs. We provide data, videos, and code of our findings in the supplementary material.

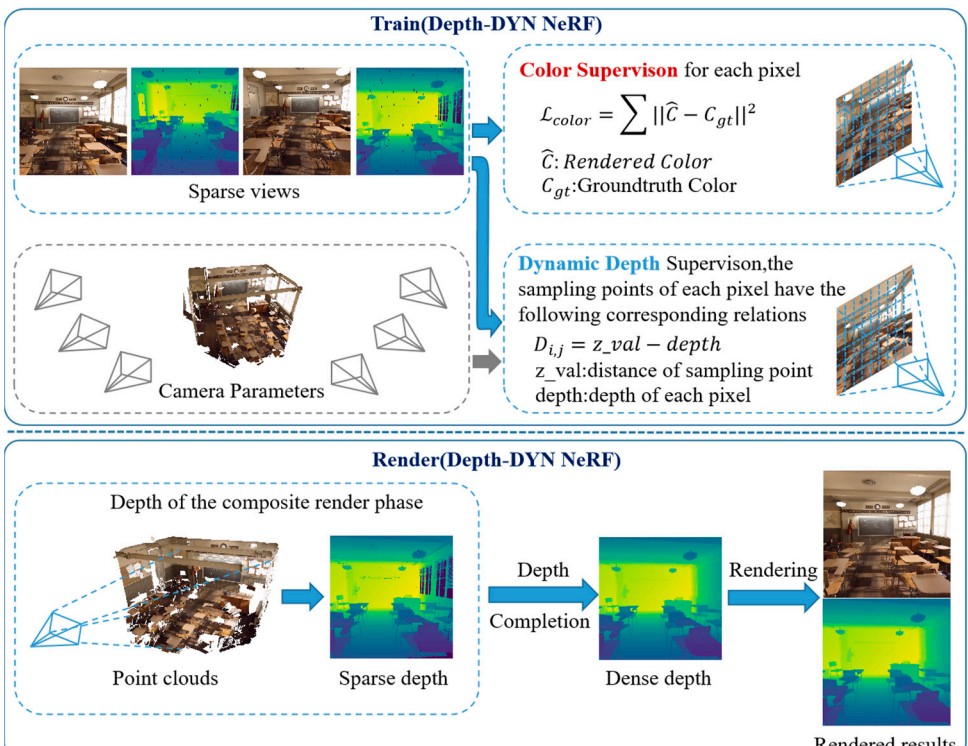

**Figure 3.** When there are few input images, the training NeRF may yield incorrect geometric structures. We use the depth information to add additional supervision to train NeRF so that each ray is sampled on the surface of the object. At the same time, we use the depth information to construct a 3D point cloud with the image, and, in the rendering stage, we obtain the depth information from the 3D point cloud to assist the rendering of NeRF. Our method is an extension of NeRF, which can be applied to any such methods.

The main contributions of our Depth-DYN NeRF are summarized as follows:

1. Depth-based Dynamic Sampling: We use additional depth supervision to train Depth-DYN NeRF. Unlike traditional NeRF, we dynamically control the spatial extent of each pixel sampled to recover the correct 3D geometry faster;
2. Depth-DYN MLP: We construct the distance constraint function between the sampled points and the depth information; then, then the distance constraint is position-encoded and input to the MLP network to guide and optimize it. As far as we know, SD-NeRF is the first algorithm that incorporates depth prior information in the MLP;
3. Depth Supervision Loss: We construct a depth-supervised loss function and a complete loss function based on the correspondence between color and depth, which effectively mitigates the appearance of artifacts, further refines the edges of objects, and shows excellent performance when multiple standard baselines are compared.

## 2. Related Work

Neural scene representation for new view synthesis. The purpose of new view synthesis is to generate images observed from a new view based on a set of captured scene images. In order to obtain the correct synthesis results, the underlying 3D geometry of the scene must be taken into account, and various scene representations are proposed for this purpose. We can obtain the new view by densely capturing the images of the observed scene using optical scene interpolation [6–8]. In the case of a small number of images, on the one hand, we can represent the scene by using its geometric information, such as texture meshes [9–12], voxels [13,14], or point clouds [15–18]. On the other hand, this could be achieved based on image-based rendering [19–23], which mainly uses a mesh model of the scene that is usually reconstructed using offline structure of motion (SfM) and multiview stereometric (MVS) methods [24–30].

Neural radiation fields with few views. NeRF-synthesizing images of new views often requires a large number of images, while NeRF for geometric reconstruction may suffer from various artifacts such as 'floaters', i.e., artifacts caused by defective density distributions, when there are too few images. Recent research has led to a decrease in the number of datasets required by NeRF in several ways. MVSNeRF [31] proposed a generalized deep neural network that can reconstruct radial fields from only three nearby input views by using fast network inference with plane-scan cost volumes for geometry aware scene inference and combing its physics-based volume rendered for neural radial field reconstruction. IBRNet [32] synthesizes a new view synthesis method for complex scenes by inserting a sparse set of nearby views, using a network architecture of multilayer perceptrons and ray transformers with radiation and volume densities at successive 5D locations. PixelNeRF [33] and metaNeRF [34] use data-driven priors recovered from the training scene fields to complete the missing information. This approach is effective when given sufficient training scenes and a limited gap between the training and test distributions; however, this assumption is not flexible enough.

Neural radiation fields with depth. Recently, there have been many algorithms that expand NeRF by adding depth prior information to NeRF, e.g., Dense Depth Priors for NeRF [35] uses a depth-complementary network running on an SfM point cloud to estimate depth in order to constrain NeRF optimization, and thus can produce higher image quality in scenes with sparse input images. DS-NeRF [36] uses the structure of motion (sfm) to generate sparse 3D points, and then the errors between the 2D key points and the projected 3D points are reprojected and the model is optimized by combining color and depth loss. DS-NeRF can train better images with fewer training views. NerfingMVS [37] uses the depth obtained using SFM reconstruction to train a monocular depth network. After this, the depth, estimated by the monocular depth network, is used to guide NeRF for learning. Finally, the quality of the depth map is further enhanced using filters based on the results of view synthesis. DoNeRF [5] proposes a dual network design to reduce the evaluation cost which carries out a depth estimation network to provide sampling locations for the coloring network by learning to solve the classification task. It also introduces a nonlinear transformation to handle large open scenes, showing that the sampling of the coloring network should occur in a distorted space to better capture the different frequencies in the foreground and background.

In our work, we created a new view synthesis technique based on a small number of samples from an implicit surface. The proposed method calculates the relationship between each sample point and depth by constructing a distance function between each ray and depth. The depth is also smoothed to deal with the edge jaggedness phenomenon, and the depth constraint is applied directly to the MLP network to help NeRF recover the geometry better. We also used a depth-complementary network for problems such as voids in the LiDAR depth data. In this way, our new view achieves high image quality and accurate depth information with a small set of inputs and sampling points. Compared to some recent algorithms that incorporate a depth prior into NeRF reconstruction, we are

able to synthesize high-quality and new views in color and depth with a small number of trained images.

## 3. Method

Our Depth-DYN NeRF differs from most of the current deep prior networks in that our Depth-DYN NeRF learns 3D scenes by combining explicit and implicit modeling. As shown in Figure 4, we propose a new network architecture that replaces the traditional 5D vector function used by NeRF to represent the scene with a 6D vector function related with a depth prior, and we approximate this continuous 6D scene display with the MLP network $F_\Theta$ and optimize its weights $\Theta$, mapping each inputted 6D coordinate to its corresponding bulk density and oriented emission colors:

$$F_\Theta(X, d, D) = (c, \sigma) \tag{1}$$

where $X$ denotes a 3D position $(x, y, z)$, $d$ is the viewing direction $(\theta, \phi)$, $D$ is the relationship function between just one point and depth information, $c$ is the output radiation (RGB color) at $X$, and $\sigma$ is the bulk density at $X$.

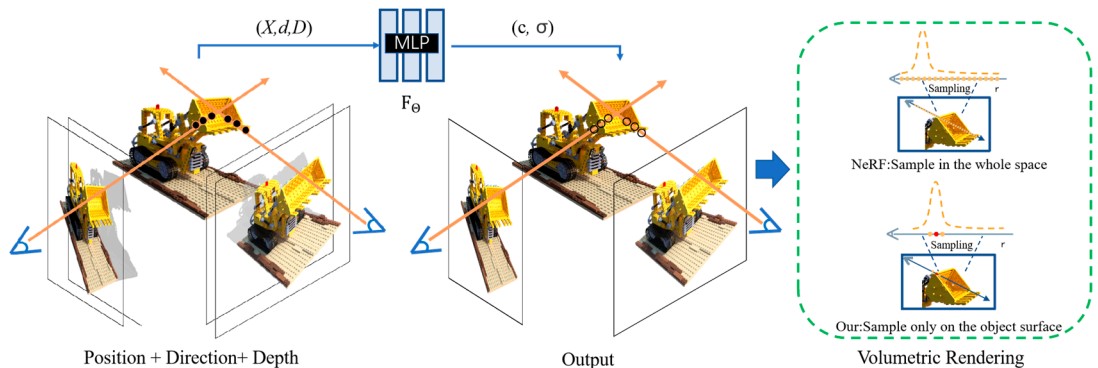

**Figure 4.** The overall framework of the Depth-DYN NeRF network, including the Depth-DYN MLP on the left and the volumetric rendering on the right.

In this paper, as shown in Figure 3, we combine RGB and depth information in the training phase to complete the training of Depth-DYN NeRF. The network structure of MLP for Depth-DYN NeRF is shown in Figure 4, where we input the position of 3D points in space, the direction of rays, and the distance function to obtain the color and voxel density of the image. In the rendering phase, we synthesize the depth data in the new view and then combine it with a depth-completion algorithm to complete the depth data. Finally, the synthesized depth is used to assist the rendering of the Depth-DYN NeRF. At all stages, the acquisition space of our rays is sampled in range based on depth.

### 3.1. Depth Completion

As shown in the rendering part of Figure 3, we use the depth-completion method to complete the depth image of the synthesized new perspective. Since there is a lot of noise in the LiDAR scanned data, particularly the depth voids in the glass material, we use the IP_Basic [38] depth-completion technique for depth completion. The depth-completion problem can be described as follows. Given a picture $I \in \mathbb{R}^{M \times N}$ and depth data $D_{sparse} \in \mathbb{R}^{M \times N}$ find $\hat{}$ that approximates a true function : $\mathbb{R}^{M \times N} \times \mathbb{R}^{M \times N} \to \mathbb{R}^{M \times N}$, where $(I, D_{sparse}) = D_{dense}$. This problem can be formulated as:

$$\min\{||\Gamma(I, D_{sparse}) - (I, D_{sparse})||_F^2 = 0\} \tag{2}$$

Here, $D_{dense}$ is the output dense depth map of the same size as I and $D_{sparse}$, and the null is replaced with the depth estimate. We achieve this by processing the operation with different sized kernels as shown in Figure 5. For a depth image as shown in Figure 6a, the

first step uses the Diamond kernel to complete the missing information next to the effective depth. The second step uses the 5 × 5 Full kernel to perform the closure operation for connecting the nearby depth values. The third step uses a 7 × 7 Full kernel to handle small to medium-sized voids. The fourth step uses a 31 × 31 Full kernel to handle the remaining larger voids, keeping the original effective pixels unchanged. The fifth step uses a 5 × 5 Cross kernel to remove the outliers presenting in the expansion process. Finally, a 5 × 5 Gaussian blur is used for smoothing. The result of the complemented depth is obtained, as shown in Figure 6b.

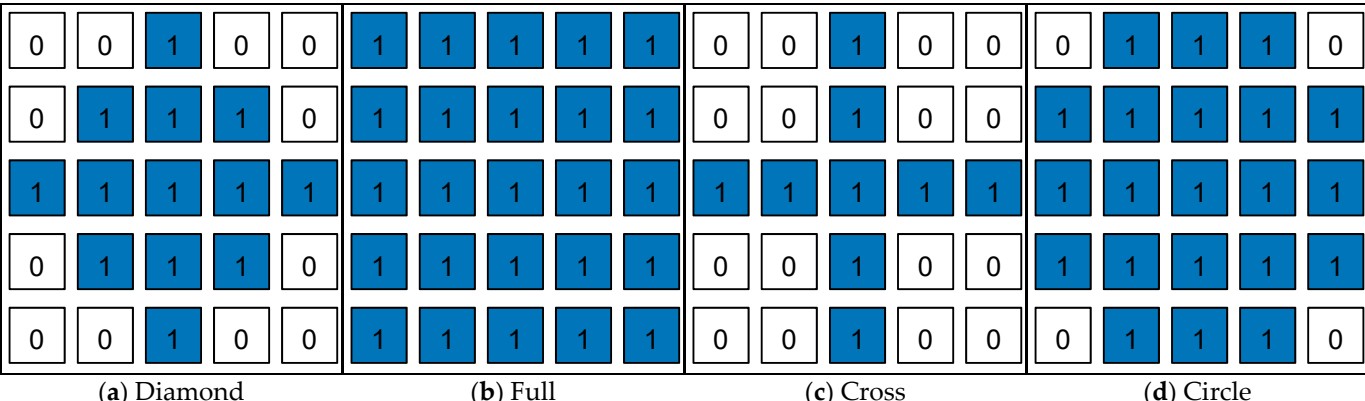

**(a)** Diamond          **(b)** Full          **(c)** Cross          **(d)** Circle

**Figure 5.** Using different sized kernels to process hollow images.

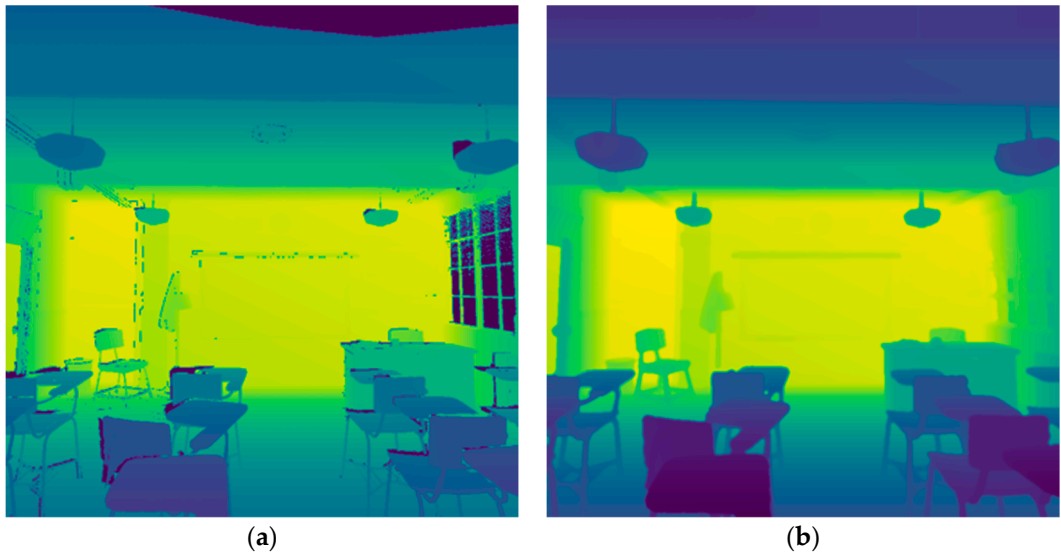

**(a)**          **(b)**

**Figure 6.** (**a**) Depth data with holes in a new perspective. (**b**) Depth recovery using complementary algorithm.

### 3.2. Depth Priors

For the distance function $D$ in Formula (1), the depth information and the distance between sampling points are input into the MLP network as constraints. As shown in Figure 7, for any pixel $P_{i,j} \in \mathbb{R}^{3 \times 3}$, we use the Level Set [39] method to construct the distance function $D_{i,j}$. $D_{i,j} = 0$ when the sampling point is on the surface of the object, and $D_{i,j}$ increases gradually between the interval [−0.5,0.5] when the sampling point is from near to far, which can be expressed as the following equation:

$$F(z\_val) = z\_val - depth, z\_val \in \Omega \qquad (3)$$

where $\Omega$ is the sample space sampling, z_val is the distance from the sample point to the camera origin in each sample space, and depth is the depth of the pixel point.

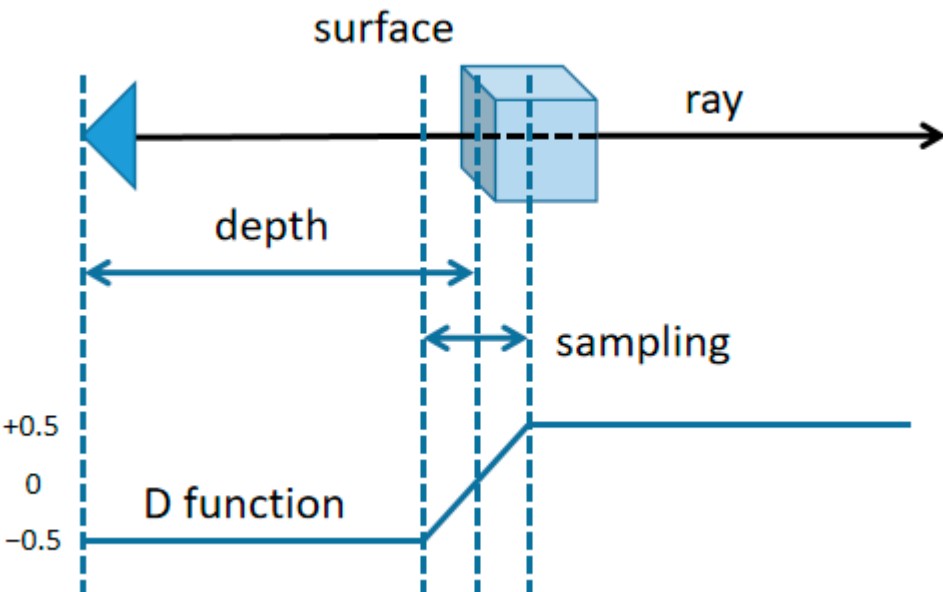

**Figure 7.** The sampling range of the rays and the variation curve of the constructed distance function are shown.

*3.3. Depth-Guided Sampling*

We propose a depth-guided dynamic sampling approach that allows the depth information to guide the rays to sample valuable parts. When we train the scene, a ray exists for each pixel. Finally, the following equation is obtained:

$$r(t) = r_o + tr_d, t \in [t_n, t_f] \tag{4}$$

where the ray origin $r_o$ is the camera center, $r_d$ is the ray direction, and $t_n$ and $t_f$ are the near and far planes of the ray.

The rays are sampled along the near and far planes between N. The location of the NeRF sampling points depends on the size of the sampling space; for large scenes, expanding the sampling space of the original NeRF may result in many invalid sampling points; moreover, a reduction in the range of the sampling space may result in distant samples not being included in the sampling space. Therefore, we propose a method to control the sampling point interval according to the depth. For each pixel P in the picture $\mathbb{R}^{M \times N}$ there exists depth values $p_i \in \mathbb{R}^{M \times N}$. We construct the depth range interval $T = [p_i - \delta, p_i + \delta]$, where $\delta$ is the hyperparameter, usually taken as 0.5. The traditional NeRF depth interval T is a constant, while our interval T is a dynamic variable. We make $t_n$ = min{T}, $t_f$ = max{T} and use this to determine the sampling interval for each pixel point. Finally, we obtain the following equation:

$$r(t) = r_o + tr_d, t \in [p_i - \delta, p_i + \delta] \tag{5}$$

*3.4. Network Training*

A joint optimization was performed for all parameters of both MLPs. A mean square error (MSE) color loss was applied for each ray r: $\mathcal{L}_{color} = [||\hat{C}(r) - C(r)||_2^2]$, where C is the ground truth color. Finally, the following equation is obtained:

$$\mathcal{L}_{color} = [||\hat{C}_c(r) - C(r)||_2^2] + [||\hat{C}_f(r) - C(r)||_2^2] \tag{6}$$

where $C(r)$ is the true RGB color, $\hat{C}_c(r)$ is the RGB color predicted by the coarse network, and $\hat{C}_f(r)$ is the RGB color predicted by the fine network. Note that, even though the final rendering comes from $\hat{C}_f(r)$, we still minimize the loss of $\hat{C}_f(r)$ so that the distribution of weights from the coarse network can be used to distribute samples in the fine network.

We add the depth loss to the loss of color as an additional constraint to the optimization problem. Here, we use the MSE depth loss:

$$\mathcal{L}_{\text{depth}} = ||\hat{Z}(r) - Z(r)||_2^2 \tag{7}$$

where $\hat{Z}(r)$ is the depth value predicted by the network and is the true depth value.

The complete loss is as follows:

$$\mathcal{L} = \sum_{r \in R} \mathcal{L}_{\text{color}}(r) + \lambda \mathcal{L}_{\text{depth}}(r) \tag{8}$$

where R is the set of rays for each batch and $\lambda$ is the hyperparameter used to balance the supervision of depth and color.

*3.5. Volumetric Rendering*

As shown in Figure 1, in the rendering stage, we construct a 3D model using forward projection of the training image based on depth information and point cloud fusion using an Iterative Closest Point (ICP) algorithm [40]. Then, the depth images are captured from the 3D model in different camera poses. As shown in Figure 6a, since the depth image has been captured from a new viewpoint, there is a large amount of empty information on the depth image, and we use the depth-complementation algorithm in Section 3.1 to perform a depth reduction operation on the empty depth image. The final restored effect is shown in Figure 6b. We restore the depth information according to Equation (2) and obtain the ray sampling space $[t_n, t_f]$ to calculate the depth and sampling point correspondence and input to the network in the adopted space according to Equation (3). In the Depth-DYN NeRF rendering process, to add additional depth, the algorithm can be constructed using the correct depth around the 3D model's construction.

In combination, MLP maps a 3D point to its color c and density $\sigma$ under the constraint of depth. To obtain an image of the new view, we need to render the color of each pixel individually. As shown in Figure 1, to render a pixel P, we emit a ray $r(t) = r_o + tr_d, t \in [t_n, t_f]$, distributing the N sampling points between $t_n, t_f$: $\{hi\}_{i=1}^N$. The color $\hat{C}$ of each pixel point p is obtained using the volume rendering formula along ray r:

$$\hat{C}(r) = \sum_{i=1}^N T_i(1 - \exp(-\sigma_i \delta_i))c_i, \text{ where } T_i = \exp\left(-\sum_{j=1}^{i-1} \sigma_j \delta_j\right) \tag{9}$$

where $\delta_i = t_{i+1} - t_i$ is the distance between neighboring samples. This function calculates $\hat{C}(r)$ from the set of $(c_i, \sigma_i)$ values is differentiable and reduces to the conventional alpha value $\alpha_i = 1 - \exp(-\sigma_i \delta_i)$.

## 4. Results

Table 1 shows the results of our evaluation, and Figure 8 shows our qualitative example. We evaluated our method using the Barbershop [5], San Miguel [5], Classroom [5] datasets. It is also compared with other methods.

**Table 1.** Quantitative comparison of the test sets of the three datasets. We use three metrics: peak signal-to-noise ratio (PSNR) (↑), structural similarity (SSIM) (↑), and image perceptual similarity (LPIPS) (↓) to evaluate the rendering quality.

| | Barbershop | | | | San Miguel | | | | Classroom | | | |
|---|---|---|---|---|---|---|---|---|---|---|---|---|
| | PSNR↑ | SSIM↑ | LPIPS↓ | Depth-MSE↓ | PSNR↑ | SSIM↑ | LPIPS↓ | Depth-MSE↓ | PSNR↑ | SSIM↑ | LPIPS↓ | Depth-MSE↓ |
| Nerf | 22.143 | 0.769 | 0.275 | 0.587 | 21.641 | 0.647 | 0.431 | 7.798 | 23.629 | 0.802 | 0.2479 | 0.575 |
| MVSNerf | 21.309 | 0.728 | 0.283 | 5.574 | 23.76 | 0.757 | 0.267 | 3.693 | 23.253 | 0.867 | 0.148 | 0.395 |
| DoNerf | 23.117 | 0.799 | 0.222 | 0.003 | 22.135 | 0.69 | 0.301 | 0.69 | 24.216 | 0.822 | 0.196 | 0.004 |
| Ours (Synthetic depth) | 24.687 | 0.843 | 0.169 | 0.0017 | 23.701 | 0.758 | 0.21 | 0.0016 | 25.188 | 0.849 | 0.161 | 0.0009 |
| Ours (ground truth depth) | 26.683 | 0.904 | 0.111 | 0.0004 | 26.148 | 0.827 | 0.17 | 0.001 | 29.059 | 0.908 | 0.096 | 0.0003 |

### 4.1. Experimental Setup

Dataset We trained on three scenes in Barbershop, San Miguel, and Classroom. They show fine, high-frequency details and large depth ranges; moreover, all datasets are rendered with blender. Each scene is trained with only 11 images; meanwhile, in order to simulate the missing depth of the LiDAR data, we perform random gouging on the depth data of the dataset. The depth is then repaired using a depth-completion network. To verify the superiority of our algorithm, we tested it from 60 new views.

NeRF Optimization In our experiments, we used 8192 rays per batch, with each coarse volume sampled at $N_c = 3$ and fine volume sampled at $N_f = 16$. The Adam optimizer was kept consistent with the original NeRF, and its learning rate started at $5 \times 10^{-4}$ and decayed exponentially to $5 \times 10^{-5}$ during the optimization process. The other Adam hyperparameters were kept at the default values $\beta_1 = 0.9$, $\beta_2 = 0.999$ and $\epsilon = 10^{-8}$. The optimization of a single scene typically requires about 100 k iterations to converge to a single NVIDIA 3080Ti GPU (about 2 to 3 h).

Evaluation Metrics Besides qualitative observations, to measure the quality of the images rendered by the network out of the new view, we also calculated the peak signal-to-noise ratio (PSNR), structural similarity (SSIM) [41], image perceptual similarity (LPIPS) [42], and mean square error (MSE) of the expected ray termination depth of the NeRF against the sensor depth to assess the accuracy of the rendered depth.

### 4.2. Baseline Comparison

We compared our method with NeRF and some recent works such as the sparse picture input MVSNeRF and DoNeRF with the addition of deep supervision. We tested it on Barbershop, San Miguel, and Classroom datasets and provided ablation experiments. The quantitative results (Table 1) show that our method outperforms the baseline criteria on all metrics.

As shown in Figure 8b, because NeRF does not add depth supervision, it is difficult to rely on RGB values to determine the corresponding relationship between images in a small number of images, resulting in a large number of "artifacts". MVSNeRF is used to learn a general network and reconstruct the radiation field across scenes from only three input images. However, due to the limited input and high diversity between different scenarios and datasets, it is impossible to achieve good results when the test scenario is too different from the pretraining scenario (example c in Figure 8). DoNeRF, on the other hand, introduces real depth information and considers only the important samples around the object surface; moreover, the number of samples required for each view ray can be greatly reduced. However, its network does not include depth information in the rendering stage, and the view differences under a small number of images are too large, resulting in the depth not being close to the true depth under the new view, which makes the image produce artifacts. Due to our depth supervision, we considered only the samples around the object and synthesized the depth information of the new view based on the depth information during the rendering phase, allowing the rays to be rendered around the samples. As shown in Figure 8, when using Barbershop, Depth-DYN NeRF greatly

reduces these artifacts compared to the baseline, with more accurate depth output and more detailed colors. Depth-DYN NeRF also improved the quality of the rendering of the detail part (Figure 8 San Miguel), where no other method could recover the detail information of the vase, and where Depth-DYN NeRF was able to clearly recover complex textures.

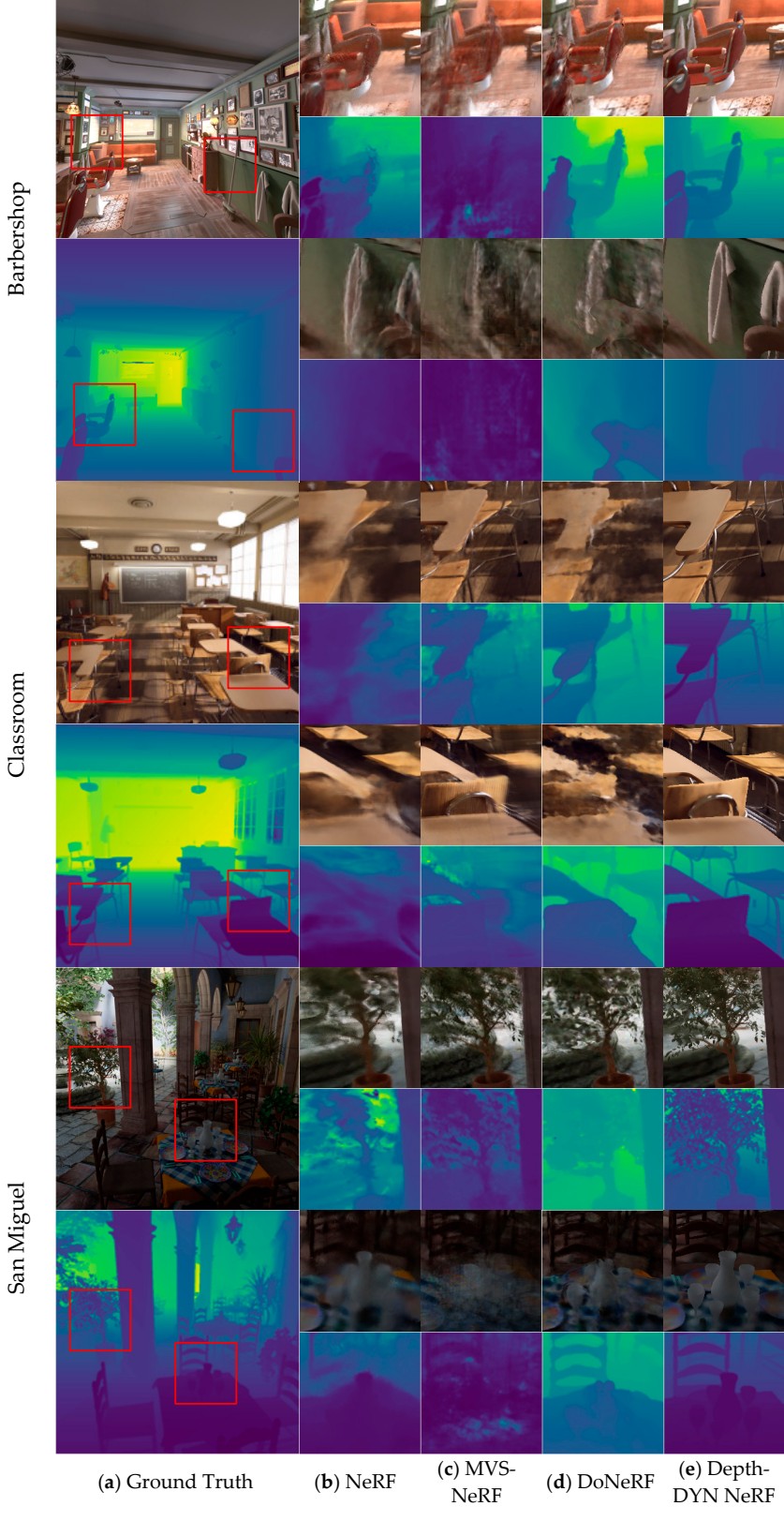

**Figure 8.** Rendered images and depths of the three test scenes vs. real images and depths.

### 4.3. Ablation Study

To verify the effectiveness of the added components, we conducted ablation experiments on the Barbershop, San Miguel, and Classroom scenes. The quantitative results (Table 2) and the rendering results (Figure 9) show that the best performance was achieved in the full version of our method in terms of image rendering as well as depth estimation. We provide full details of the experiments in Appendix A.

**Table 2.** Ablation studies performed on our model. The table shows a quantitative comparison of Depth-DYN NeRF without depth loss (w/o DPLOSS), without depth loss and depth MLP (w/o dploss, Depth-DYN MLP), and without dynamic depth and depth MLP (w/o dysamp, Depth-DYN MLP).

| | Barbershop | | | | San Miguel | | | | Classroom | | | |
|---|---|---|---|---|---|---|---|---|---|---|---|---|
| | PSNR↑ | SSIM↑ | LPIPS↓ | Depth-MSE↓ | PSNR↑ | SSIM↑ | LPIPS↓ | Depth-MSE↓ | PSNR↑ | SSIM↑ | LPIPS↓ | Depth-MSE↓ |
| Depth-DYN NeRF | 26.683 | 0.904 | 0.111 | 0.0004 | 26.148 | 0.827 | 0.17 | 0.001 | 29.059 | 0.908 | 0.096 | 0.0003 |
| w/o DPLOSS | 26.378 | 0.899 | 0.115 | 0.009 | 25.919 | 0.83 | 0.158 | 0.713 | 28.188 | 0.904 | 0.098 | 0.033 |
| w/o dploss, Depth-DYN MLP | 24.937 | 0.869 | 0.1455 | 0.028 | 25.963 | 0.8198 | 0.182 | 0.003 | 27.48 | 0.88 | 0.1287 | 0.0008 |
| w/o dysamp, Depth-DYN MLP | 20.475 | 0.694 | 0.396 | 0.251 | 21.656 | 0.595 | 0.46 | 2.021 | 23.852 | 0.776 | 0.281 | 0.111 |

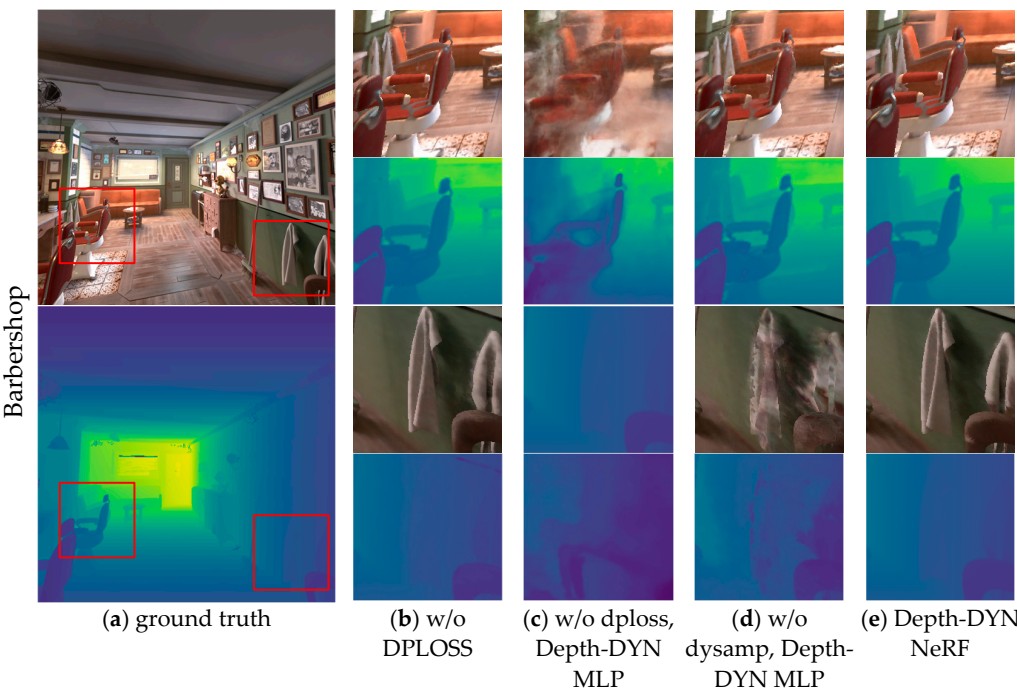

(**a**) ground truth  (**b**) w/o DPLOSS  (**c**) w/o dploss, Depth-DYN MLP  (**d**) w/o dysamp, Depth-DYN MLP  (**e**) Depth-DYN NeRF

**Figure 9.** Rendered image and depth of the test scene vs. real image and depth.

w/o dysamp, Depth-DYN MLP In this experiment, we only added the depthloss module to the original NeRF and observed a significant improvement in the depth of the model during rendering but not in the quality of the rendered image.

w/o dploss, Depth-DYN MLP In this experiment, we only used the dynamic sampling space module to make changes to the way NeRF is sampled; the MLP input and the loss function are used in the original NeRF method.

w/o dploss, In this experiment, only the Depth-DYN MLP was used as well as the dynamic sampling space, without adding depth loss. It is observed that the generated depth images are not particularly accurate in some parts, especially in sharp areas.

Depth-DYN NeRF In this experiment, we used the Depth-DYN MLP; dynamic sampling space and depth loss modules to render high-quality image levels and reconstruct better results at edges and complex textures.

*4.4. Depth-DYN NeRF with Synthetic Depth*

Our validation of the Depth-DYN NeRF rendering of the scene with synthetic depth (ours (synthetic depth)) (Figure 10) shows an average loss of about 3 dB compared to the Depth-DYN NeRF (see Table 1). Classroom has the largest loss; for the barbershop, the difference between using synthetic depth and using real depth is the most minimal. The texture recovery is also excellent, especially in the near details. For San Miguel (Figure 10), Depth-DYN NeRF was able to reconstruct the details of the leaves, and the rendering results show that, although we used synthetic depth for the rendering of the image, Depth-DYN NeRF (synthetic depth) still produced sharper results than DoNeRF.

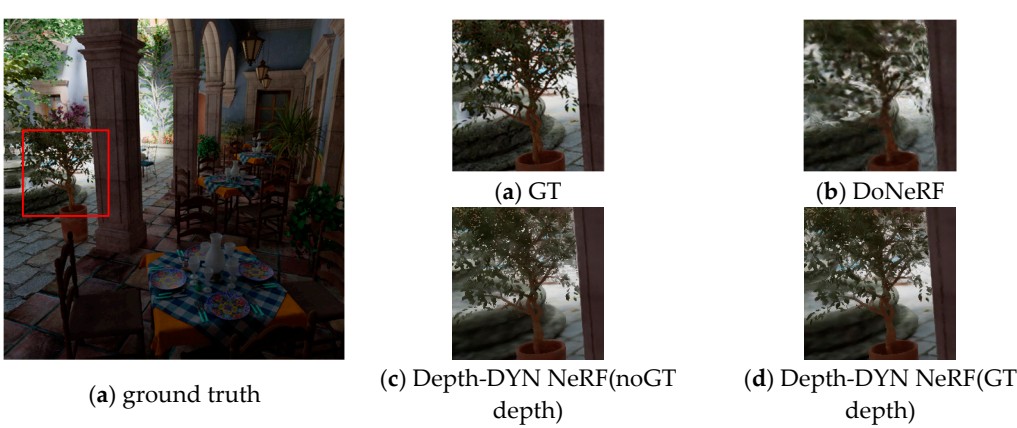

(**a**) ground truth     (**a**) GT     (**b**) DoNeRF

(**c**) Depth-DYN NeRF(noGT depth)     (**d**) Depth-DYN NeRF(GT depth)

**Figure 10.** Results of synthetic depth rendering.

**5. Limitations**

Although our method is an improvement over the baseline method, there may be various problems with depth in real scenes, and accurate depth images may not be recovered in depth recovery. Therefore, Depth-DYN NeRF may only provide authentic results for data with fewer depth voids.

*Future Work*

In this paper, we only focused on static scenarios, and will consider dynamic NeRF algorithms in the future. By using depth data to constrain and guide the optimization, we combine spatiotemporal data to build dynamic 3D scenes.

**6. Conclusions**

We propose a new attempted synthesis method using a depth prior to the neural radiation field, which uses RGB-D data to learn the neural radiation field and obtain a better geometric mechanism as well as a faster training time. We can use fewer views to train the neural radiation field because the deep supervision provides additional supervision with efficient sampling. The use of a deep prior enables the neural radiation field to learn more accurate geometric information using limited views. Experiments have shown that our method outperforms the state-of-the-art methods both quantitatively and qualitatively by learning accurate geometric mechanisms in 10 to 20 training images.

**Supplementary Materials:** The following supporting information can be downloaded at: https://github.com/joeyw1030/depth-DYN-NeRF, accessed on 10 January 2023.

**Author Contributions:** Conceptualization, J.W. and J.X.; methodology, J.W.; software, J.W.; validation, J.X., X.X. and X.Z.; formal analysis, J.W.; investigation, J.W. and X.Z.; resources, J.W.; data curation, J.W.; writing—original draft preparation, J.W.; writing—review and editing, X.Z.; visualization, J.W.; supervision, J.X.; project administration, J.X. and X.Z.; funding acquisition, T.J. and Z.J. All authors have read and agreed to the published version of the manuscript.

**Funding:** This work was supported in part by the Ningbo Science and Technology Innovation 2025 Major Project (Grant #: 2022Z077, 2021Z037), and the Key R&D Project of the National Emergency Management Department under Grant 2021XFCX352025.

**Institutional Review Board Statement:** Not applicable.

**Informed Consent Statement:** Not applicable.

**Data Availability Statement:** The data are not publicly available due to privacy.

**Acknowledgments:** We would like to thank the editors and the anonymous reviewers for their insightful comments and constructive suggestions.

**Conflicts of Interest:** The authors declare no conflict of interest.

## Appendix A

We will provide more details about the datasets we used. We conducted single-scene learning experiments on three datasets.

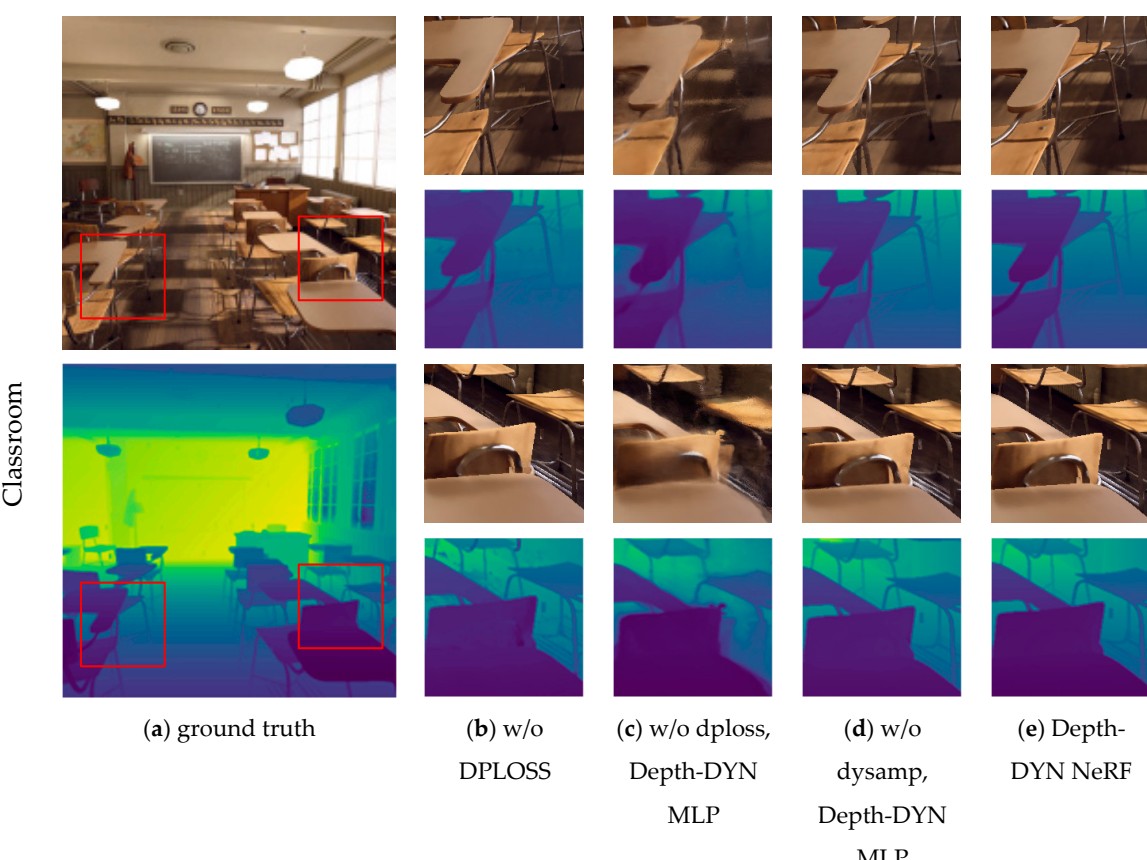

(**a**) ground truth    (**b**) w/o DPLOSS    (**c**) w/o dploss, Depth-DYN MLP    (**d**) w/o dysamp, Depth-DYN MLP    (**e**) Depth-DYN NeRF

**Figure A1.** The rendered image of the Classroom scene and the depth of the test scene are compared with the real image and depth.

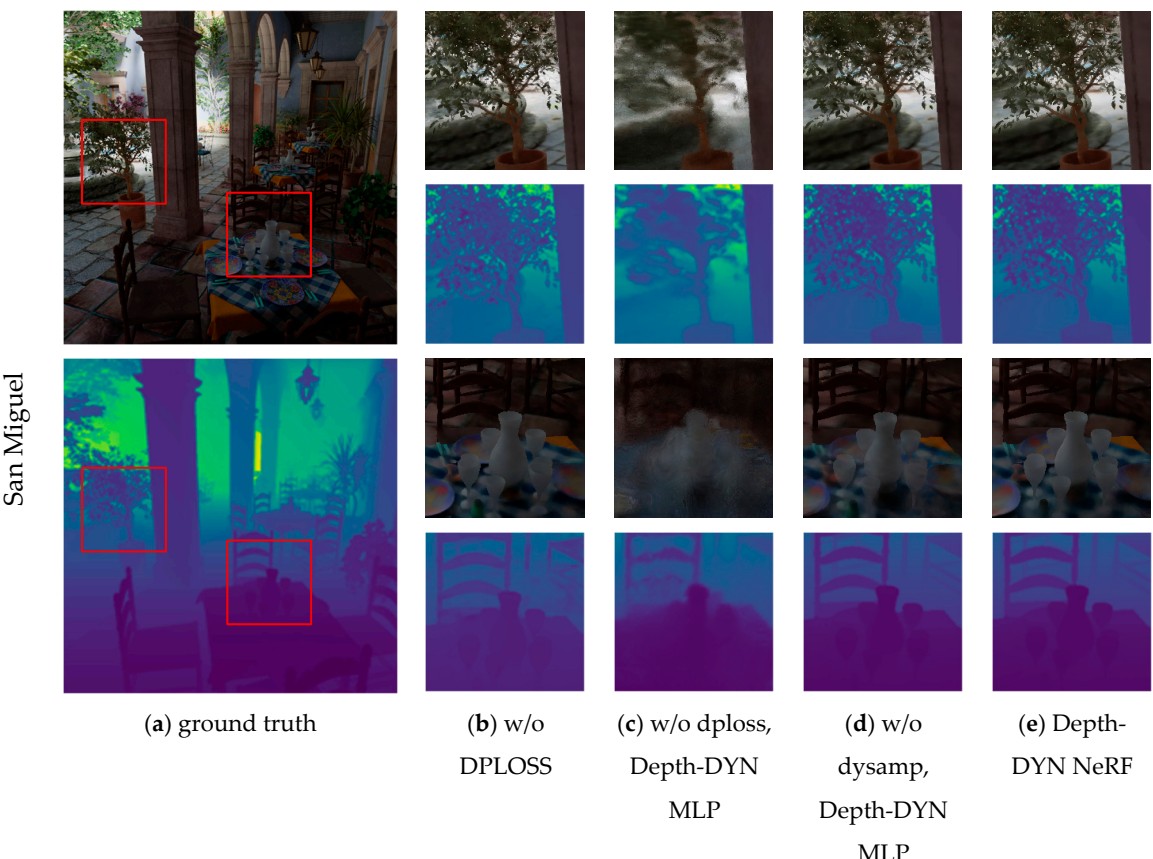

| (**a**) ground truth | (**b**) w/o DPLOSS | (**c**) w/o dploss, Depth-DYN MLP | (**d**) w/o dysamp, Depth-DYN MLP | (**e**) Depth-DYN NeRF |

**Figure A2.** The rendered image of the San Miguel scene and the depth of the test scene are compared with the real image and depth.

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
