# Peer review of "Depth-Based Dynamic Sampling of Neural Radiation Fields"

_electronics, doi:10.3390/electronics12041053_

Round 1
Reviewer 1 Report
This work proposes to incorporate depth into NeRF reconstruction for new view synthesis. It is based on relating each sample point with the depth, and the advantage is that a small number of input views are required.
This paper needs a major revision. Novelty of this work needs to be elaborated. My concerns are:
a) Section 3 is difficult to read. There is need of clarity in describing the flow of the proposed scheme.
b) Many of the math symbols and notations are not consistent/accurate.
c) Eq. 5 is same as 4.
d) Line 100: sentence is incomplete.
e) There is no D_dense in Eq. 2 as referred on line 185.
f) Line 247: What is ICP?
g) Line 251: Fig. 7 —> Fig. 6(b) ?
Author Response
Point 1:Section 3 is difficult to read. There is need of clarity in describing the flow of the proposed scheme.
Response 1: We thank the reviewer for raising this question. I modified Figure 4 to better explain the third chapter, and I modified the contents of lines 172-182 to explain the third chapter.
Point 2: Many of the math symbols and notations are not consistent/accurate.
Response 2: We thank the reviewer for raising this question. I modified the mathematical conformity of lines 172, 188, 189, 204, 236, 237, 238, 239, 267, 268 and formula 2.
Point 3: Eq. 5 is same as 4.
Response 3: We thank the reviewer for raising this question. I changed formula 5 to .
Point 4: Line 100: sentence is incomplete.
Response 4: We thank the reviewer for raising this question. Corrected at line 100.
Point 5: There is no D_dense in Eq. 2 as referred on line 185.
Response 5: We thank the reviewer for raising this question. On line 190, check D_ Dense explained.
Point 6: Line 247: What is ICP?
Response 6: We thank the reviewer for raising this question. I added references to ICP in 252.
Point 7: Line 251: Fig. 7 —> Fig. 6(b) ?
Response 7: We thank the reviewer for raising this question. I made corrections on line 193 and line 201, Figure 6 (a) and Figure 6 (b) respectively.

Reviewer 2 Report
Good work done. If possible try to add some sentences about how different is your method from the interferometry methods.
Author Response
Point 1:Good work done. If possible try to add some sentences about how different is your method from the interferometry methods.
Response 1: We thank the reviewer for raising this question. In lines 307 - 327, I explained in detail the interpretation of the results between different algorithms and the advantages of our algorithm.

Reviewer 3 Report
Overall this paper has been well written and coherent, in which the contribution of the Depth-DYN NeRF method can be presented properly. Unfortunately, some parts need to be explained in more detail. Some suggestions that need to be done to improve the presentation of this paper, namely:
1. In the abstract, the contribution of the method needs to be emphasized with a numerical/percentage value of the improvement.
2. It is better if the layout of the Figure is adjusted to the order in which it is explained, especially in Figure 1 it should be placed after Fig. 3.
3. Fig. 4 needs to be more adapted to the stages in section 3
4. It would be better if the methods compared in Table 1 were given no citation
5. In the results section, the explanation is not yet detailed. You need to provide a deeper analysis of the results presented and what the effect of modifying the method you are proposing is. Both regarding the rendered image and depth.
Author Response
Point 1:In the abstract, the contribution of the method needs to be emphasized with a numerical/percentage value of the improvement.
Response 1: We thank the reviewer for raising this question. Added the description of relevant values in line 20.
Point 2: It is better if the layout of the Figure is adjusted to the order in which it is explained, especially in Figure 1 it should be placed after Fig. 3.
Response 2: We thank the reviewer for raising this question. I changed the position of the picture according to the order mentioned in the picture, and moved Figure 1 to Figure 2.
Point 3: Fig. 4 needs to be more adapted to the stages in section 3
Response 3: We thank the reviewer for raising this question. I modified Figure 4 to better explain the third chapter.
Point 4: It would be better if the methods compared in Table 1 were given no citation
Response 4: We thank the reviewer for raising this question. I added a reference on line 271.
Point 5: In the results section, the explanation is not yet detailed. You need to provide a deeper analysis of the results presented and what the effect of modifying the method you are proposing is. Both regarding the rendered image and depth.
Response 5: We thank the reviewer for raising this question. In lines 307 - 327, I explained in detail the interpretation of the results between different algorithms and the advantages of our algorithm.

Round 2
Reviewer 1 Report
Authors have incorporated my comments adequately.
However, English language proofreading is needed.